# Marine Antithrombotics

**DOI:** 10.3390/md18100514

**Published:** 2020-10-13

**Authors:** Rohini Dwivedi, Vitor H. Pomin

**Affiliations:** Department of BioMolecular Sciences, Pharmacognosy Division, Research Institute of Pharmaceutical Sciences School of Pharmacy, University of Mississippi, Oxford, MS 38677-1848, USA; rdwived1@olemiss.edu

**Keywords:** alkaloids, anticoagulation, antithrombosis, fucosylated chondroitin sulfate, polyphenols, sulfated polysaccharides, terpenoids

## Abstract

Thrombosis remains a prime reason of mortality worldwide. With the available antithrombotic drugs, bleeding remains the major downside of current treatments. This raises a clinical concern for all patients undergoing antithrombotic therapy. Novel antithrombotics from marine sources offer a promising therapeutic alternative to this pathology. However, for any potential new molecule to be introduced as a real alternative to existing drugs, the exhibition of comparable anticoagulant potential with minimal off-target effects must be achieved. The relevance of marine antithrombotics, particularly sulfated polysaccharides, is largely due to their unique mechanisms of action and lack of bleeding. There have been many investigations in the field and, in recent years, results have confirmed the role of potential marine molecules as alternative antithrombotics. Nonetheless, further clinical studies are required. This review covers the core of the data available so far regarding the science of marine molecules with potential medical applications to treat thrombosis. After a general discussion about the major biochemical steps involved in this pathology, we discuss the key structural and biomedical aspects of marine molecules of both low and high molecular weight endowed with antithrombotic/anticoagulant properties.

## 1. Introduction

Cardiovascular diseases are a major contributor to the global disease burden. Behind most cardiovascular diseases, such as myocardial infarction, ischemic stroke and coronary artery disease, lies a pathological condition—thrombosis [1,2]. Thrombosis is a diseased state which involves the formation of a blood clot in the vessels of the cardiovascular system. Severe morbidities, including fatal ones, can be caused when the blood flow in the vessels is occluded by the presence of blood clots or thromboses. An estimated 18 million deaths are attributed to pathologies related to thrombosis every year [3]. A pathological blood clot formation in the arteries is called an atherothrombosis, while in the veins this is known as a venous thrombosis. An arterial thrombus is mainly composed of platelets, while a venous thrombus is primarily composed of fibrin and red blood cells, giving it the appearance of red clots. Sometimes, a venous blood clot dislodges from its origin and mobilizes to a secondary location and is known as an embolus. It has usually been observed to move to the lungs, giving rise to a potentially fatal pulmonary embolism (PE). Venous thrombosis along with PE is termed venous thromboembolism (VTE), which accounts for 60,000–100,000 global deaths every year [4,5].

There are various risk factors, including acquired and genetic ones, which predispose a person to developing thromboses [6,7]. The development of a thrombus is primarily attributed to the damaged endothelial lining of vessels [5], the hypercoagulable state, and arterial and/or venous blood stasis [4]. However, there are important differences in the pathophysiological mechanism underlying the development of a venous or arterial thrombus [8]. For example, platelet activation plays a more important role in the case of arterial thrombosis compared to venous thrombosis. Arterial thrombosis is generally caused by the disruption of an atherosclerotic plaque, leading to the recruitment of platelets and consequently causing adhesion, aggregation and eventual thrombus formation [5].

The process begins when inactivated platelets from the blood become adhered to exposed collagen at the site of the ruptured vasculature (Figure 1). Attachment to collagen is accomplished directly via the glycoprotein receptor (GP) VI and GP Ia/IIa or by binding to a von Willebrand factor (vWF) using surface glycoprotein receptor GP Ib/V/IX. The adhesion of platelets leads to their activation, causing conformational change, the release of molecular contents (from alpha and dense granules) and, most importantly, the activation of GP IIb/IIIa. While the granular contents of the platelets—adenosine diphosphate (ADP), fibrinogen, thromboxane A_2_ (TxA_2_), vWF, etc (Figure 1)— activate more platelets, the activated GP IIb/IIIa causes platelet aggregation by fibrinogen mediated bridging of the two platelets [9,10,11]. Continued platelet accumulation leads to the formation of platelet rich thrombus at the injured site. The exposed tissue factor further initiates the coagulation cascade. The coagulation system moves into a series of activation steps of the various coagulation factors, leading to the generation of fibrin which, in turn, forms a mesh and stabilizes the thrombus (Figure 2). The solidified mass of aggregated platelets with fibrin is then completed in the formation of the arterial thrombus [12,13].

Venous thrombosis, unlike arterial thrombosis, is not initiated by vascular injury. It is generally predisposed to occur in regions like deep veins, where endothelial cells lining the venous wall express adhesion molecules in response to various factors such as decreased flow, hypoxia, etc. As the rate of blood flow slows in the venous lumen, the amount of oxygen is decreased, and the blood concentration is increased. This situation creates a hypercoagulable and hypoxic environment in the vessel which decreases the expression of certain anticoagulant proteins while increasing the expression of procoagulants. The expression of procoagulants like P-selectin promotes attachment of tissue factor expressing immune cells (like leukocytes) to the endothelium. The tissue factor then triggers a coagulation cascade, again leading to the formation of a fibrinous rich blood clot in the venous vessels [4,8,14].

Obstructed blood vessels due to pathological thrombotic events lead to poor blood flow and, consequently, manifest into severe conditions including ischemic heart disease, stroke, myocardial infarction and limb ischemia [3]. The signs, symptoms and severity of a thrombosis vary depending on the location of the thrombus and on various other molecular/cellular factors which determine the treatment regimen to be followed in a subject specific manner.

### 1.1. Current Antithrombotics

Thrombosis is clinically treated with antithrombotic drugs. These molecules prevent the formation of thrombus by inhibiting platelet aggregation (antiplatelet), as in the case of arterial thrombosis, or by preventing the formation of a fibrin strand (anticoagulant), as in VTE. Once the thrombus is formed, fibrinolytic or thrombolytic drugs are generally used to dissolve the clots [15]. Anticoagulant drugs include the widely known and prescribed heparin, warfarin, dabigatran and rivaroxaban [8]. Heparin is administered parenterally, while warfarin, dabigatran and rivaroxaban are orally administered drugs. Low molecular weight heparin (LMWH) and unfractionated heparin (UFH) are indirect thrombin inhibitors because they act by accelerating the serpins, such as antithrombin (AT) and heparin cofactor-II (HC-II), in the inhibition of thrombin (Figure 2). Direct oral anticoagulants, such as rivaroxaban, apixaban and edoxaban, work by inhibiting FXa. Dabigatran is a direct thrombin inhibitor capable of inhibiting thrombin even when it occurs in its bound form with fibrin. Warfarin, a well-known vitamin K antagonist, prevents coagulation by inhibiting the C1 subunit of the vitamin K epoxide reductase enzyme, consequently down-regulating the synthesis of clotting factor [16]. Figure 3 illustrates the mechanism of action of the commonest antiplatelet drugs.

Platelets have different surface receptors, for example, GP Ia/IIa, GP VI, GP Ib/V/IX, GP IIb/IIIa, P_2_Y_12_, TxA_2_, the purinergic G protein-coupled receptor (P_2_Y) and the protease-activated receptor 1 (PAR 1), for binding to various agonists including collagen, vWF, fibrinogen, ADP, TxA_2_, and thrombin. These receptors, when activated by binding to their respective ligands, lead to platelet adhesion, activation and aggregation, as described above. General antiplatelet drugs function by targeting these specific surface receptors to prevent pathological arterial thrombosis (Figure 3) [8]. These drugs can be classified into cyclooxygenase (COX) inhibitors, ADP receptor antagonists, PAR 1 inhibitors and GP IIb/IIIa inhibitors, depending on the receptor they are targeting. Aspirin, one of the oldest and most common of the antiplatelet drugs prescribed, belongs to the class of COX inhibitors. This effect works by inhibiting the COX 1 enzyme, which is necessary for the synthesis of TxA_2_ from arachidonic acid. Aspirin irreversibly acetylates the hydroxyl group of a serine residue of the COX 1 enzyme. Selective acetylation of this serine prevents the binding of arachidonic acid to the enzyme and stops its conversion to TxA_2_. Due to its efficacy, aspirin has been therapeutically used to prevent primary and secondary episodes of VTE, even though its role is mainly towards arterial thrombosis. The inhibition of platelet activation can also be accomplished by inhibiting the P_2_Y_12_ receptor, a major ADP receptor on platelets [17]. Drugs like clopidogrel, prasugrel, ticagrelor and ticlopidine are used clinically as ADP receptor antagonists. Vorapaxar, a PAR 1 inhibitor, prevents platelet activation by inhibiting the thrombin-mediated cleavage of PAR 1. Platelet aggregation is primarily triggered by the activation of GP IIb/IIIa receptor. Inhibitors like eptifibatide, tirofiban and abciximab, which specifically target this receptor, prevent the binding of fibrinogen to the GP IIb/IIIa surface receptor and consequently reduce platelet aggregation and thrombus growth [8,17].

### 1.2. Challenges Associated with Current Antithrombotics

Despite the availability of several antithrombotics (listed above), an extensive search for novel potential drugs is in progress due to the severe limitations associated with current therapies available. Bleeding is the major side effect of antithrombotics, causing mild to severe hemorrhage with potentially fatal consequences [3]. This makes regular monitoring during the treatment procedure absolute necessary. However, vulnerable people at high risk, for example, old people or people with renal dysfunction, sometimes succumb to these undesirable side effects despite proper clinical managements [18,19]. This either requires termination of the therapeutic effect by antidote, requiring constant patient monitoring, or administration of the drugs in systemic low doses, leading to undesirable treatment outcomes. Heparin, the most widely used anticoagulant, presents challenging therapeutic effects due to its associated high risk of bleeding as a side effect. Its low bioavailability and parenteral mode of administration also contributes to its limitations. One of the major drawbacks of heparin, which challenges its usage in several cases, is heparin-induced thrombocytopenia (HIT) (Figure 4).

HIT is the most common example of an antithrombotic drug-induced severe pathological consequence and it has a high mortality rate if not diagnosed in time. The disease generally becomes symptomatic from 5 to 14 days after heparin treatment. The alpha granules of the activated platelet release a positively charged molecule—platelet factor 4 (PF 4)—which, along with heparin forms the PF 4-heparin immunogenic complex, triggering the formation of immunoglobulin G (IgG) antibodies. IgG antibodies bind to the PF 4-heparin complex, forming a PF 4-heparin-IgG complex. This ternary complex then binds to platelets via the FCγRIIa receptor and activates them. The activated platelets release prothrombotic microparticles, which in turn promote the release of thrombin. More PF 4 is then released due to this activation, causing further antibody and heparin production. Meanwhile, binding of the heparin-IgG complex to heparan sulfate activates the endothelial layer, increasing the production of tissue factor. The released tissue factor drives the generation of more thrombin, thus making the body prone to thrombosis. These HIT events trigger the immunological system in a way that the body becomes more prone to further blood clots, thereby increasing the severity of the pathology [20].

Besides heparin, other existing drugs also have severe clinical limitations. Oral anticoagulants, such as warfarin, cause severe intracranial hemorrhage, as it is capable of crossing the placenta if administered during pregnancy, leading to fetal hemorrhage [21]. Drugs like tirofiban, which works by blocking GP IIb/IIIa receptors, cause permanent inhibition to platelet aggregation, and this leads to significant bleeding. In cases of dual antiplatelet therapy (DAPT) involving administration of aspirin and clopidogrel, it has been observed that the risk of bleeding can increase by up to 1.8-fold in DAPT compared to aspirin-based treatments, although this is less risky than treatment involving aspirin and warfarin together [3]. Direct oral anticoagulants (DOACs) are thought to exert less bleeding than vitamin K antagonists, such as warfarin, but the bleeding risk can increase when DOACs are administered along with aspirin. DOACs-therapy seems to minimize the bleeding risk when administered alone, but does not completely eliminate the risk of fatality [3,22]. This makes the usage of potent anticoagulation antagonists indispensable in the case of DOAC therapy as well. Idarucizumab and andexanet alfa are approved intravenous drugs capable of reversing DOAC effects, while another drug—ciraparantag—is under clinical trial. Just a few antidotes are currently available: Idarucizumab is the only reversal drug against dabigatran, while andexanet alfa reverses the effect of apixaban and rivaroxaban [23,24]. Moreover, the observed interaction of andexanet alfa with tissue factor pathway inhibitor and the possible immunogenicity of both drugs does raise concerns about their potential side effects [23,25]. In addition, the production of both drugs by recombinant technology increases their costs, posing a practical limitation to their usage [26]. The limited availability of risk free and cost-effective anticoagulant reversal drugs further increases the challenges for current anticoagulant therapy. Together, these circumstances help to drive support for research to discover new anticoagulants.

Combined therapies of antiplatelets and anticoagulants with constant dose adjustments are used to obtain the maximal benefit of these drugs while avoiding clinical complications [3]. None of the existing antithrombotic medications, however, are free from side effects, and thus require regular and continuous monitoring of the subject. With the limitations of the present antithrombotics, the development of a potent and safe antithrombotic is still needed in the field of thrombotic research [27]. To this end, efforts are being made across the globe to design, synthesize and identify potent antithrombotic molecules from various natural sources. Due to their unique properties and immense significance, molecules from marine sources have been reviewed frequently. A recent comprehensive review of marine antithrombotics by Carvalhal et al. in 2019, covered polysaccharides and small molecules from the ocean which exhibit anticoagulant activity [28]. This review will cover similar and updated investigations that have been performed on novel marine antithrombotic molecules over the past five years and commented on the structure and associated anticoagulant activities of molecules including alkaloids, polyphenols, terpenes, peptides, sulfated glycans and sulfated rhamnans. We also include some antithrombotic studies from the past five years related to lipids, alginate and chitosan. These substances were not covered in Carvalhal et al [28].

## 2. Marine Antithrombotics

The marine environment harbors different species that consist of a diverse repertoire of therapeutic molecules. Subject to different environmental conditions than terrestrial organisms, marine organisms produce chemically and structurally unique molecules/metabolites as an adaptive strategy to survive in water bodies. These chemically novel molecules (peptides, sulfated polysaccharides, small molecules, etc.) have significant bioactivities with established therapeutic implications (antithrombotic, antimicrobial, antitumor, antifungal, antiviral, anticancer properties, etc.) [29,30,31]. In the quest of identifying new antithrombotics capable of preventing thrombus formation without disturbing hemostasis, molecules from marine organisms are emerging as a promising solution. Many recent publications have appeared describing methods for the isolation and the partial or full structural characterization of novel molecules from marine sources endowed with potential antithrombotic activity. Although these studies need further exploration to be taken forward clinically, they provide many potential molecules for rational manipulation with a view to obtaining targeted antithrombotics in the future.

### 2.1. Organic Small Molecules

An estimation of 13,000 chemicals have already been extracted and isolated from marine sources [32]. These metabolites include molecules like polyphenols, steroids, polyketides, terpenoids, fatty acid and alkaloids. The associated chemical and structural diversity of these molecules makes them uniquely functional. Many of these molecules have been proven to be potent antithrombotics.

#### 2.1.1. Polyphenols

Many polyphenolic compounds have been extracted thus far from marine sources including phlorotannins, flavanoids, lignins, and tannins. These are accredited with several medicinal properties [33]. Phlorotannins, which are chemically oligomers of phloroglucinol, are structural components of brown algae (Figure 5A). The extract of brown alga *Eisenia bicyclis* (rich in phlorotannins) was recently shown to exhibit antithrombotic activity when examined in vivo and in vitro. Characterization of the phlorotannin extract led to the identification of the following components present in its chemical composition: 6,6′-bieckol, 6,8′-bieckol, 8,8′-bieckol, dieckol and phlorofucofuroeckol A. In an arteriovenous shunt rat model of thrombosis, the *E. bicyclis* extract caused a decrease in thrombus formation by 14% and 36.6% at a concentration of 100 mg/kg and 200 mg/kg of body weight, respectively. The extract was also found to cause clot retraction in vivo. The antiplatelet activities of the extract were evident from the reduction in ADP stimulated platelet aggregation in a dose dependent manner, with almost 90% inhibition observed at a dose of 12.5 µg/mL. The phlorotannin-rich extract could inhibit ADP stimulated calcium mobilization completely at 12.5 µg/mL. The inhibited adenosine triphosphate released from the platelet granules can decrease P-selectin expression along with inhibition of fibrinogen binding (~100% at 25 µg/mL) to GPIIb/IIIa. The study postulates that the brown algal extract (rich in polyphenols) is capable of executing an antithrombotic effect via impairing P_2_Y_12_ signaling [34].

#### 2.1.2. Sphingosines

Sphingosine, an amino alcohol, is an important constituent of sphingolipids (Figure 5B). A functional study was carried out to extend the bioactive role of sphingosines isolated from the marine sponge *Haliclona tubifera* [35]. The link between cancer and a predisposition to thromboses is well reported. Increased expression of tissue factor has been adjudged to be one of the crucial reasons for the hypercoagulable condition associated with cancers. The study aimed to elucidate the role of sphingosines in cancer and coagulation pathways. The results, besides confirming the antitumor activity of sphingosines, which was the focus of the investigation, also commented on the potential anticoagulant activity exhibited by them against human citrated plasma samples, as concluded from the recalcification studies [35]. Sphingosines exhibited a recalcification time of 9 min at the highest concentration of 100 μg/mL.

#### 2.1.3. Terpenes

Terpenes are hydrocarbons comprised chiefly of isoprene units. Terpenes and terpenoids from marine sources are known for their therapeutic roles including anti-inflammatory, anti-microbial, anti-angiogenic and anti-cancer activity [36]. In an earlier study, three diterpenes isolated from the marine brown alga *Dictyota menstrualis*—pachydictyol A (Figure 5E), isopachydictyol A, and dichotomanol—were shown to inhibit thrombin activity directly. Dichotomanol was reported to inhibit 50% of thrombin activity at an examined concentration of 0.35 mM, while pachydictyol A and isopachydictyol A could inhibit it to the same extent at a concentration of 0.68 mM [37]. A molecular modelling-based mechanistic investigation of these diterpenes was carried out to understand how they inhibit thrombin, as the derived information could also be helpful in designing similar ligands for thrombin inhibition in the future. In silico studies demonstrated the potential binding site of diterpenes to the catalytic site of thrombin, through which the molecules could cause the catalytic inefficiency of this enzyme. Dichotomanol, with the highest AT activity, had the highest dipole moment as well, which plausibly accounts for its good coulombic/electrostatic interactions within the thrombin active site [38]. Frondoside, a triterpenoid isolated from sea cucumber *Cucumaria frondosa*, is known to inhibit the phosphatidylinositol 3-kinase (PI3K) pathway in cancer cells. Like its properties in platelet activation and aggregation, frondoside has also shown antithrombotic effects. Frondoside was able to reduce the conformational activation of GPIIb/IIIa platelet receptor in a dose dependent manner, which has a significant role in platelet aggregation. It also caused a potential decrease in thrombus formation, evidenced by the prolongation of the vessel occlusion time in a mouse model, while it increased tail bleeding [39].

#### 2.1.4. Benzoic Acid Derivative

*R*-/*S*-2-(2-Hydroxypropanamido) benzoic acid (*R*-/*S*-HPABA) is a small organic molecule (Figure 5D) with anti-inflammatory and analgesic properties [40]. It was first isolated from the fermentation broth of marine fungus *Penicillium chrysogenum*. Upon investigating its role in platelet aggregation, it was found to exhibit inhibition of ADP-induced platelet aggregation. In vivo studies demonstrate its potential in causing substantial recovery from collagen epinephrine induced pulmonary thrombosis (Table 1). Moreover, this benzoic acid derivative has COX 1 inhibitory potential, which seems to be stereoselective, since the S isomer was found to be more efficient than the R isomer. The study proposed that the antithrombotic activity of the molecule was somewhat correlated with the inhibition of COX 1 activity [40].

#### 2.1.5. Alkaloids

Marine organisms like sponges, alga, and tunicates are known to produce alkaloids. Indole alkaloids of all these organisms have been assigned various medicinal roles [41]. Fascaplysin, an indole alkaloid isolated from the Fijian marine sponge *Fascaplysin opsi*s (Figure 5C), is a kinase inhibitor and its potential antithrombotic role was investigated. The role of fascaplysin in down regulating the PI3K pathway in cancer cells inspired this study, as this pathway is involved in platelet activation as well. In a photochemically induced thrombus in a mouse model, treatment with fascaplysin prolonged the obstruction time in the vessel. Moreover, it also increased the tail vein bleeding time apart from inhibiting the platelet aggregation. The observed inhibition of platelet aggregation was due to a decreased activation of GPIIb/IIIa upon fascaplysin treatment [42].

### 2.2. Biomacromolecules

#### 2.2.1. Lipids

Phospholipids (PL), the basic constituent of almost all cell membranes, are composed of chains of fatty acids, glycerol and phosphate groups. These amphiphilic molecules isolated from marine sources have been assigned with many promising therapeutic properties including their cardioprotective ability [43]. The predominance of omega 3 polyunsaturated fatty acids (ω3 PUFA) has been found to play a major role in the cardiovascular system. However, its therapeutic properties are a synergistic result of several other fatty acids and their ratios are not just limited to the PUFA itself. Tsoupras et al. investigated the role of phospholipids isolated from *Salmon salar* against platelet activating factor (PAF) pathway for the first time [44]. Salmon phospholipids (PL) could inhibit PAF-stimulated aggregation of platelets at low doses but were only able to prevent thrombin-induced platelet aggregation at higher doses (Table 2). This suggests that inhibition mediated through PAF is the major mechanism of antithrombotic action.

ω3 PUFAs were found to be predominant in the salmon PL preparation and their content was higher than that of omega 6 (ω6) fatty acids. The critical ratio of ω6/ω3 PUFA was found to be approximately 1/2.5, which probably endows the potential cardioprotective function to the extracted PL [44]. Changing the process of extraction of phospholipids from conventional extraction (CE) to food grade extraction (FGE) changed the observed activity of salmon extracted polar lipids. Food grade extracted PLs exhibited lower inhibitory effects towards the PAF pathway than CE PL extracts. FGE-extracted salmon PLs displayed increased inhibitory effects for thrombin-stimulated human platelet aggregation (Table 2). Phosphatidylethanolamine and phosphatidylcholine, both components of the extracted PLs, were found to be potent anti-PAF and exhibited high AT activity [45].

In another report by the same group, salmon heads (SHs), brain, eyes and main optic nerves (SBEON), head-remnants after SBEON removal, herring fillets, herring heads and minced boarfish (MB) were examined for the first time to evaluate their probable antithrombotic activity [46]. The extracted phospholipids were tested for their effect on platelet aggregation mediated by PAF, thrombin, collagen and ADP. The results demonstrated the potential inhibitory effect of PLs on platelet aggregation induced by the mentioned small molecules. As a comparison, the results showed that PL from salmon head could strongly inhibit (IC_50_ = ~90 µg) thrombin-induced platelet aggregation, those from SH (~IC_50_= ~180 µg) and SBEON (~IC_50_ = ~100 µg) exhibited the highest potency against ADP-induced platelet aggregation, while MB PLs (~IC_50_ = ~50 µg) strongly inhibited collagen-induced platelet aggregation. PAF-induced platelet aggregation was inhibited by the PLs from all fractions similarly. The active component isolated from all of these sources was found to be richest in PUFA, followed by monounsaturated fatty acids and saturated fatty acids [46].

#### 2.2.2. Peptides

Marine organisms are a good source of proteins and their digested peptides. These linear chains of amino acids present strong therapeutic prospects due to their small size and wide-ranging bioactivity. Due to their biomedical activities, many of these marine peptides are sold as nutraceuticals on the market and many are currently undergoing clinical trials [47]. Apart from their antioxidant, antihypertensive, antitumor, antiviral, antibacterial properties, research investigations have also shown that they show antithrombotic properties.

Nori *Porphyra yezoensis* is a widely cultivated seaweed in Japan. Although it contains from 25 to 50% protein, the bioactivity of its proteinaceous content has not been extensively explored [48]. A recent study reported anticoagulant activity by obtaining pepsin hydrolyzed fractions of Nori protein. Multiple fractionation and purification of the pepsin hydrolysate of Nori protein led to the identification of a hexadeca-peptide anticoagulant peptide, with the sequence characterized as NMEKGSSSVVSSRMKQ. The peptide is characterized as having a post translational modification of methionine, converting it to methionine sulfoxide. The presence of charged residues possibly renders this peptide with anticoagulant activity, as has been seen in the case of many anticoagulant peptides. Peptide exhibits a dose-dependent increase in activated partial thromboplastin time (aPTT), with a concentration going from 0.15 μM to 3 μM corresponding to an aPTT of approximately 120 to 320 sec, respectively, while an untreated control exhibited an aPTT of 30 sec. The prothrombin time (PT) and thrombin time (TT), however, remained unaffected at these concentrations. Prolongation of the aPTT time compared to PT suggests that the mode of action was carried out by targeting intrinsic coagulation pathway [48].

In another study, tissue from oyster *Crassostrea gigas* was hydrolyzed to generate peptides with possible antithrombotic activity. The study is relevant as it used an in silico strategy and was validated with biological assays to establish the antithrombotic activity of oyster peptides. The peptides were obtained by post gastral digestion and were characterized by mass spectroscopic methods; they were found to exhibit sequence similarity to the known thrombin inhibitors hirudin, bivalirudin and tsetse thrombin inhibitor. These were then investigated by molecular docking to elucidate their mechanism of interaction with thrombin. The 14 different peptides obtained were able to prolong aPTT, PT and TT timings as well as inhibit thrombin. aPTT, TT and PT increased to 59.90 sec, 29.05 sec and 18.28 sec compared to the control (34.62 sec, 13.13 sec and 16.13 sec respectively) at a concentration of 2 mg/mL. The digested peptides exhibited a 40% inhibition of thrombin activity upon treatment. Peptides were inferred to function by affecting the conformation of the thrombin active site and diminishing its activity. The probable mechanism of antithrombotic activity was demonstrated by docking the peptide LSKEEIEEAKEV, which had the highest affinity for thrombin. Similar to hirudin, this peptide works as a competitive thrombin inhibitor by preventing the binding of fibrinogen at exosite I, owing to the negative charge of the peptide [49]. In another report, *C. gigas* tissue was digested with different proteases—pepsin, papain, trypsin and neutral protease. The most active digested fractions were those of pepsin hydrolysate. The aPTT and TT exhibited by pepsin hydrolysate were more prolonged than other protease hydrolysates, except for neutrase hydrolysate, which showed a similar TT to that of pepsin (Table 3). Further purification of the active fraction led to the isolation of the peak corresponding to the peptide sequence TARNEANVNIY (Figure 6), namely, CAGP, which prolonged the aPTT from 10.5 sec to 14.2 sec at a concentration of 250 μg/mL. Similar to hirudin, the mechanism for CAGP peptide and thrombin interaction was suggested to entail CAGP competing with fibrinogen for exosite I [50].

Blue mussel *Mytilus edulis* is a marine mollusk known to harbor many bioactive compounds. Trypsinization of the sarcoplasmic proteins, myofibrillar proteins and matrix proteins of this mollusk yielded 387 unique peptides. The sarcoplasmic protein, myofibrillar protein and matrix protein fractions exhibited antithrombotic activity of 40.17%, 85.74%, 82%, respectively, at a concentration of 5 mg/mL. ELEDSLDSER, obtained upon digestion, was characterized to be the peptide with the highest affinity for thrombin based on the score obtained upon docking [51]. In another report on *Mytilus edulis*, proteins were isolated from different parts of the mollusk (i.e., foot, byssus, pedal retractor muscle, mantle, gill, adductor muscle, viscera) and digested enzymatically to obtain various peptides. Based on the bioactivity predictor database, the digested peptides obtained from adductor muscle were examined for anticoagulant activity. Three peptides—VQQELEDAEERADSAEGSLQK, RMEADIAAMQSDLDDALNGQR, and AAFLLGVNSNDLLK—exhibited high antithrombotic activity [52]. Tachyplesin I, a well-known antimicrobial peptide, was examined for its antithrombotic effects in vitro and in vivo. It was found to increase the bleeding time and clotting time (CT). It also inhibited platelet aggregation and thrombosis by interfering with the PI3K/protein kinase B (AKT) pathway. Due to its low toxicity and targeted action, it has potential to be developed as a novel antithrombotic drug [53].

#### 2.2.3. Proteins

Protein-based protease inhibitors are spread widely across plant and animal kingdom and have garnered attention due to their therapeutic applications. Their potential application as an anticoagulant has been explored in few studies. In one recent study, a trypsin inhibitor isolated from marine bacterium strain *Oceanimonas sp*. BPMS22 was shown to increase the clotting time from 7 to 20 min in an in vitro analysis. The extent of the observed coagulation was 33% less than the control [54]. Another study reported that a serine protease isolated from *Sipunculus nudus* (a marine worm) exhibited in vitro fibrinolytic activity by activating plasminogen. Its antithrombotic effect was also seen in FeCl_3_-induced carotid arterial thrombus in rats [55]. Thus, proteases are a candidate to be screened for their antithrombotic potential.

#### 2.2.4. Sulfated Glycans

The ocean is a source of some specific sulfated glycans that have significant anticoagulant and antithrombotic properties. Sulfated polysaccharides found in marine organisms include glycosaminoglycans (GAGs) such as chondroitin sulfate (CS), dermatan sulfate (DS), heparin, heparan sulfate (HS) and fucosylated chondroitin sulfates (FucCS); and GAG-like molecules including sulfated fucans (SFs) and sulfated galactans (SGs). The sea is also a source of other polysaccharides like alginates, sulfated rhamnan and chitins, etc. These polysaccharides are widely distributed among marine organisms including seaweed, fish, sea cucumbers, marine algae, shrimps, sea urchins, etc. The high negative charge imparted via sulfation makes these polysaccharides capable of interacting with proteins/factors involved in critical biological processes like coagulation, thus endowing them with significant antithrombotic properties. The interest in marine GAGs has also been generated following the issues of contamination with the GAGs isolated from terrestrial sources, causing severe toxic consequences. We attempt to systematically cover the relevant reports of novel antithrombotics in this class of marine sulfated glycans.

##### CS/DS

CS is composed of alternating β-d-glucuronic acid (GlcA) and *N*-acetyl β-d-galactosamine (GalNAc) within disaccharide units, while DS has the GlcA replaced by α-l-iduronic acid (IdoA). CS is further classified into CS-A, CS-B (DS), CS-C, CS-D and CS-E based on the sulfation pattern and uronic acid of the repeating disaccharides units (Figure 7).

The extraction of GAGs from the bone of the fish *Sciaena umbra* was followed by its extensive compositional analysis and characterization studies. The corb bone GAG (CBG) comprises CS and DS chains at a 3 to 1 ratio. The presence of disulfated disaccharide units in the backbone of the CS/DS mixture from bony fishes is the reason for their anticoagulant activity, which is why the sulfated CBG was examined for its in vitro anticoagulation activity by aPTT, PT and TT assays. CBG, at a concentration of 1000 μg/mL, was found to cause a 2.6-fold prolongation of clotting by aPTT and 1.2-fold increase in PT time compared to the control. When tested for TT, CBG showed a 3.5-fold prolongation compared to the control. Thus, CBG shows an anticoagulation effect mediated by both an intrinsic and an extrinsic coagulation cascade. The presence of disulfation in the disaccharide at IdoA/GlcA2S and GalNAc4S, or at IdoA/GlcA4S and GalNAc6S, appeared to corroborate the inhibitory thrombin activity of this marine GAG [56].

GAGs were also extracted from the skin of the corb of *Sciaena umbra*. The isolated corb skin glycosaminoglycan (CSG) (Figure 7) exhibited promising anticoagulant activity mediated via extrinsic and intrinsic coagulation pathways. CSG was also characterized to be a CS/DS mixture, which exhibited a predominant mono 4-sulfation and di-2,4-sulfation of the GalNAc residue. CSG differed from CBG in terms of the ratio of CS/DS. Unlike CBG, in CSG the CS was found to be around 25%, while the content of DS was 75% [57]. In a subsequent in vivo study carried out on rats to assess the anticoagulant activity of both, it was shown that the administration of CBG and CSG at 25 and 75 mg/kg of body weight could increase the aPTT and TT values. However, the PT prolongation that was reported by in vitro studies could not be observed in the in vivo model [58].

##### Heparin/HS

Heparin/HS has chains comprising repeating disaccharide units of GlcA/IdoA and *N*-acetyl α-d-glucosamine (GlcNAc)(Figure 8). Despite being the most celebrated anticoagulant clinically, usage of heparin is not free from side effects. Thus, studies investigating heparin-like molecules from marine source with minimal risk are ongoing in the field. A novel shrimp-derived hybrid heparin/HS molecule (i.e., sH/HS) was isolated from the head of *Litopenaeus vannamei* and exhibited aPTT prolongation and anti-FXa activity (97% inhibition at about 0.6 μg/mL) with negligible bleeding risk. The sH/HS was reported to consist of disaccharide units containing GlcNAc6S linked to GlcA, N-sulfated glucosamine linked to IdoA2S, and GlcNAc6S linked to IdoA2S [59]. As FIIa and FXa are shown to be important for the induction of tumorigenic events, molecules with anti-FXa and anti-FIIa are considered potential antitumorigenic compounds. Adriana et al., while attempting to correlate the anti-FIIa and antitumor activity of novel sH/HS, reported its anti-FIIa activity. Their study demonstrated that the GAG could inhibit FIIa activity by 90.7% at a minimal dose of 0.5 μg/mL. It was also shown that sH/HS can induce HS production by endothelial cells upon treatment (Table 4) [60]. Importantly, the sH/HS does not carry a risk of bleeding at the same concentration (100 μg/mL) which porcine intestinal mucosa heparin (used as control) was shown to cause significant bleeding in 2 min [59]. A potent sulfated GAG was also reported to be isolated from cuttlefish *Sepia pharonis*. Comprised of GlcA and GlcNAc, this polysaccharide also exhibited prolonged aPTT and PT of 91 IU and 39.55 IU at 25 μg/mL, respectively, demonstrating its potential anticoagulant activity [61].

##### FucCS

Fucosylated chondroitin sulfates (FucCS) are unique marine GAGs with a CS-like backbone decorated with fucosyl branches. The backbone of FucCS primarily consists of 4-linked GlcA and 3-linked GalNAc units, having sulfated α-l-fucopyranosyl (Fuc*)* branches linked to the O3 position of GlcA (Figure 9). Due to their ability to target the intrinsic tenase complex of the coagulation cascade, they are looked upon as prospective anticoagulant molecules. Since the discovery of this class of sulfated polysaccharides in 1988, they have been continuously isolated from various species of sea cucumber [62]. Reports in the past five years show an extensive interest in investigation of this class of sulfated glycans. We are covering the reports presenting isolation of new FucCS from other species of sea cucumber. Their reported anticoagulant activities are summarized in Table 5.

Among the three polysaccharides isolated from the sea cucumber *Patallus mollis*, one was characterized to be a FucCS called PmFG. It was shown to comprise a CS-E backbone. CS-E consists of sulfation at both the C4 and C6 positions of GalNAc of the repeating disaccharide unit. Extensive structural elucidation led to the characterization of this structure as {-(l-FucR-α-1,3)-d-GlcA-β-1,3-d-GalNAc4S6S-β-1,4-}_n_. The sulfation pattern of fucose branches was found to be Fuc2S4S, Fuc3S4S and Fuc4S at a ratio of 2:2.5:1, represented as R in the structure. The determination of its anticoagulant activity revealed that PmFG did not affect PT but was able to prolong TT and aPTT. The concentration required to prolong aPTT was less than that observed for LMWH and equivalent in the case of TT (Table 5). The polysaccharide also exhibited anti-factor Xase (FXase) activity [63]. Two FucCS were reported from *Holothuria coluber* by Yang et al. The polysaccharide in one case was found to comprise a classical CS-like backbone with α-1,3 linked fucose branches with four sulfation patterns: Fuc2S4S, Fuc3S, Fuc3S4S and Fuc4S. This FucCS could not affect PT but was able to prolong aPTT and TT activity twice as much as LMWH. Consistent with FucCS reports from other sea cucumbers, it was able to inhibit intrinsic FXase with an IC_50_ of 14.73 ng/mL [64]. The second FucCS obtained from *H. coluber* was also capable of inhibiting FXase; it differed structurally to the previous FucCS by its lack of Fuc3S sulfation pattern [65].

The structure of FucCS isolated from *Apostichopus japonicas* has been previously reported by various research groups; however, there were some differences in the structure reported from each characterization in terms of the CS backbone type, fucosyl branching and sulfation patterns. Guan et al. tried to depolymerize the polysaccharide and, using the bottom-up strategy approach of mass spectrometry, tried to readdress the structural investigation. The intensive structural elucidation strategy confirmed the *A. japonicas* FucCS to have a CSE-like backbone. The position of the fucose branches was assigned and exclusively linked to GlcA with the possible sulfation patterns reported as Fuc2S4S, Fuc3S4S or Fuc4S. The native FucCS prolonged the aPTT time and showed a 10-fold higher anti-FXase activity than LMWH. Depolymerized fragments of this glycan also exhibited an aPTT prolongation and anti-FXase activity, although the effect was smaller than that of native sugar, signifying the effect of chain shortening on the function. The requirement of a nonasaccharide, in terms of chain length, appeared to be necessary for the anticoagulant activity of *A. japonicas* FucCS [66]. Fucosylated glycan (FCS_hm_) reported from *Holothuria mexicana* was found to bear a unique fucosyl branch attached to the O6 of GalNAc, besides the regular fucosyl unit linked to the O3 position of GlcA. The sulfation pattern of the fucose attached to GlcA was shown to be Fuc2S4S or Fuc4S, while fucose linked to GalNAc had Fuc4S and Fuc3S4S sulfation patterns. This glycan was active in anticoagulation and also inhibited AT-mediated anti FIIa and anti-FXa functions [67]. A FucCS known as H_m_G also has potent activity, as reported from a study involving *H. Mexicana*. It predominantly has FuC4S branches linked to GlcA of its CS-C type backbone [68].

The *Stichopus hermanni*-derived FucCS was also capable of targeting intrinsic tenase complex, and was found to have Fuc2S4S, Fuc4S and Fuc3S4S branches. However, the integration of the peaks suggested Fuc2S4S to be the major sulfation pattern; thus, the structure was considered to be {-(l-Fuc2S4S-α-1,3)-d-GlcA-β-1,3-d-GalNAc4S6S-β-1,4-}_n_ [69]. *Holothuria pollii* FucCS exhibited anti FIIa activity mediated by HC-II and AT. This FucCS, bearing 2S4S and 3S4S sulfation patterns, was found to exhibit a procoagulant effect at low concentrations ranging from 0.05 to 0.005 µg/mL, but caused usual antithrombotic activity above 10 µg/mL [70]. A highly regular FucCS was isolated from *Massinium magnum* (Figure 9). Its structure was unique as it only comprised Fuc3S4S branches. Its backbone structure was primarily found to be {-(l-Fuc3S4S-α-1,3)-d-GlcA-β-1,3-d-GalNAc4S6S-β-1,4-}_n_, with minor fractions of {-(l-Fuc3S4S-α-1,3)-d-GlcA-β-1,3-d-GalNAc4S-β-1,4-}_n_. The anticoagulant activity assessed for this sugar is shown in Table 5 [71]. A comparative study of FucCSs derived from two sea cucumbers (*Cucumaria frondosa* and *Thelenota ananas*) suggested that sulfation degree and molecular weight seem to be driving factors affecting the anticoagulant activity of these glycans, while the sulfation pattern seems to play a relatively inferior role [72]. Apart from the above mentioned reports, FucCS isolation and structure function characterization have also been carried out for *Bohadschia argus* [73], *Holothuria scabra* [74], *Cucumaria japonica* [75], *Holothuria fuscopunctata* [76], *Holothuria lentiginosa* [77] and *Pearsonothuria graffei* [78], all of which exhibited potential anticoagulant activities mediated via targeting the intrinsic coagulation pathway (Table 5).

The potential anticoagulant activityof FucCS, established to be mediated by serpin-dependent and serpin-independent mechanisms, are severely challenged by their associated tendencies of Factor XII (FXII) activation and platelet aggregation. This limitation is a major hurdle for the prospective journey of FucCS on the clinical pathway. Many structural activity studies aim to decrease the side effects of this unique sulfated GAG through chain length manipulation. Depolymerization of *Isostichopus badionotus* FucCS has been found to decrease aPTT, TT, anti-FXa, and anti FIIa/AT activity; however, this also led to decreased FXII activation [79]. Although a minimum length of the polymer has been considered a prerequisite for anticoagulant activity, the strategy of depolymerization has been proven to enhance selectivity to a large extent.

**Table 5 marinedrugs-18-00514-t005:** Anticoagulant activity measurements of FucCS reported from sea cucumbers in recent years.

Sea Cucumber Spp.	aPTT	TT	PT	Anti FXase	Anti FIIa/Plasma	Anti FXa/AT	Anti FXa/HCII	Anti FIIa/AT	Anti FIIa/HCII	Mol. Wt	Ref.
	**μg/ml**			**IC _50_ (ng/mL)**						**kDa**	
*Apostichopus japonicas*	3.06	-	-	9.20	-	-	-	-	-	76.4	[66]
*Bohadschia argus*	4.13	-	1280	14.83	-	3341	-	530.8	-	70.3	[73]
*Cucumaria frondosa*	-	-	-	-	-	1000	-	500	1000	58	[72]
*Cucumaria japonica*	2.5	-	-	-	-	-	-	-	-	-	[75]
*Holothuria coluber*	4.94	10.32	1280	14.73	-	-	-	-	-	49.48	[64]
*Holothuria coluber*	3.31	7.68		26.4	-	10000	-	1260	-	54.9	[65]
*Holothuria fuscopunctata*	3.45	6.46	1280	41.9	-	10000	-	448	589	42.6	[76]
*Holothuria lentiginosa*	30 IU/mg	-	-	-	10.2	5.5	-	0.7	-	50.8	[77]
*Holothuria mexicana*	100	150	-	-	-	1000	-	100	-	-	[67]
*Holothuria pollii*	220 IU/mg	-	-	-	-	-	-	125	35	45.8	[70]
*Holothuria scabra*	20	60	-	-	-	-	-	-	-	69.1	[74]
*Massinium magnum*	2.8	6	-	-	-	-	-	-	-	27	[71]
*Patallus mollis*	3.5	10.7	-	13.7	-	-	-	-	-	60.3	[63]
*Pearsonothuriagraffei*	20.9	9.84	-	330	-	5490	-	5080	-	73	[78]
*Thelenota ananas*	-	-	-	-	-	1000	-	500	1000	63	[72]

PT-Prothrombin time.

##### GAG-Like Molecules

GAG-like molecules include SFs and SGs. These sulfated homopolysaccharides, primarily isolated from marine sources like algae, are known for having complex structures associated with their heterogeneity. SFs and SGs isolated from echinoderms are much more regular than those obtained from algal sources, and their high yield, low bleeding risk and efficient anticoagulant activity make them of special interest. Table 6 lists the anticoagulant activity of some of the GAG-like molecules described below.

##### Sulfated Galactans

Sulfated galactansare linear chains of 3-linked β-d-galactopyranose and 4-linked α-d/l-galactopyranose or 3,6-anahydro-α-d-galactopyranose, sulfated at specific positions (Figure 10). The isolation of an SG from the Brazilian ascidia *Microcosmus exasperatus* is the first report of a galactose-based polymer from this tunicate. Structurally, it comprises 4-linked α-l-galactopyranose units with sulfation in some residues at the 3-position. This molecule is capable of doubling the aPTT time at a concentration of 20 µg/mL [80]. *Codium isthmocladum*-derived two 3-linked-β-d-SGs with branching at the C6, which exhibited 1.7-fold prolonged CT compared with the standard at a concentration of 10µg/mL. These polysaccharides, besides galactose and sulfate, also show a presence of pyruvate groups at the O3 and O4 positions, adding uniqueness to the structure of these SG molecules [81]. Two SGs from *Udotea flabellum* were found to exert anticoagulant activity similar to heparin. These compounds were, however, incapable of inhibiting thrombin directly but could inhibit it via AT [82]. The SGs purified from red seaweed *Spyridia hypnoides* also showed promising anticoagulant activity, with an aPTT of 25.36 IU and a PT of 2.46 IU at a concentration of 25 µg/mL. Structurally, it was found to consist of 3-linked-β-d-galactopyranose unit and 4-linked-3,6-anhydro-α-l-galactopyranose as the repeating units [83].

##### Sulfated Fucans

Like SGs, SFs—also known as fucoidans in brown algae—are polymeric chains of repeating sulfated Fuc units (Figure 11). Primarily isolated from marine algae and sea cucumber body walls, these sulfated polysaccharides exhibit a different heterogeneity depending on the source they are isolated from. Branched xylofucan sulfate isolated from the brown alga *Punctaria plantaginea* was converted to its partially and highly sulfated analogs. The highly sulfated SF analogs, when examined for anticoagulant and antithrombotic properties, showed a prolongation of clotting by aPTT. They also inhibited platelet aggregation induced by ristocetin. Partially sulfated and desulfated SF-derived analogues from this alga were found to be functionally inactive. Sulfation was found to be critical in the AT-mediated inhibition of thrombin or FXa [84]. The SF NP2, isolated from *Nemacystus decipiens*, was found to have a backbone comprising 3-linked Fuc residues and a branch composed of Fuc-(2→1)-GlcA. The paper reports promising antithrombotic prospects of NP2 due to its high fibrinolytic activity [85].

Besides FucCS, sea cucumber body walls are also rich in SFs. SFs isolated from the body wall of *Patallus mollis* had a backbone comprised of {-l-Fuc2S-α-1,4-}_n_ repeating units. The backbone had branches bearing Fuc4S and Fuc3S sulfation patterns. Like other SFs, the *Patallus mollis* SF (known as PmFS) also exhibited aPTT prolongation and intrinsic tenase inhibitory potential [63].

**Table 6 marinedrugs-18-00514-t006:** Anticoagulant activity measurements of GAG-like molecules from marine organisms.

Source	Species	aPTT	TT	PT	Anti FXase	Anti FXa/AT	Anti FIIa/AT	Anti FIIa/HCII	Mol. Wt	Ref
		**μg/ml**			**IC _50_ (ng/mL)**				**kDa**	
***Green alga***	*Udotea flabellum*	3	-	-	9.20	-	500	-	76.4	[82]
***Brown alga***	*Punctaria plantaginea*	>100	>100	1280	14.83	no	no	-	70.3	[84]
***Sea cucumber***	*Holothuria albiventer*	25.79	115.47	>1280	71.99	-	-	-	>2000	[86]
	*Holothuria coluber*	78.92	>1280	>1280	244	-	-	-	64.55	[64]
	*Holothuria fuscopunctata*	11.3 IU/mg	-	-	92.8	1780	882	2947	36.8	[87]
	*Holothuria pollii*	2.5	2	-		-	125	35	45.8	[88]
	*Patallus mollis*	24.3	-	-	74	-	0.5	0.16	6.12	[63]
	*Stichopus horrens*	19.6 IU/mg	-	-	51.2	53256	3758	-	487.9	[87]
	*Thelenota ananas*	10.4 IU/mg	-	-	196.7	1150	1176	292.6	61.2	[87]
***Sea Urchin***	*Lytechinus variegatus*	-	-	-	-	0.29 IU/mg	0.44 IU/mg	-	-	
	*Strongylocentrotus franciscanus*	-	-	-	-	0.05 IU/mg	0.02 IU/mg	-	-	[89]
	*Echinometra lucunter*	-	-	-	-	0.27 IU/mg	0.56 IU/mg	-	-	

An SF derived from *Holothuria coluber* that displayed anticoagulant activity was identified to have a backbone of repeating tetrafucose units linked via α-1,4 linkages. This SF also presents Fuc4S side chains attached to its backbone [64]. Potent inhibitory action against the intrinsic tenase complex was also shown by three SFs isolated from *Holothuria fuscopunctata*, *T. ananas* and *Stichopus horrens*. The repeating units were identified as (l-Fuc3*S*-α-1,4−)_n_, -(l-Fuc2*S*-α-1,4−)_n_, (l-Fuc2*S*-α-1,3−)_n_ for SFs from *H. fuscopunctata*, *T. ananas* and *S. horrens*, respectively; they provide a unique model to understand and correlate the anticoagulant effect of these SFs to sulfation pattern and position of glycosidic linkage [87]. Another SF isolated from the sea cucumber *Holothuria albiventer* comprised of regular α-1,3 linked hexasaccharide repeating units of Fuc rings with potential sulfation at O3, O3,4, O2,3 or O2,3,4 positions. This polysaccharide exhibited aPTT and TT prolongation while displaying intrinsic tenase activity. Depolymerization of this SF led to a decrease in anticoagulation activity, as reported earlier with different classes of sulfated glycans [86]. An SF with a backbone comprising 3-linked α-L-Fuc residues and with sulfation at the 2-position was obtained from the sea cucumber *Stichopus horrens* with an aPTT value of 3.92µg/mL [90].

The inhibition of platelet aggregation was observed by an SF extracted from *Holothuria polii*. The SF was characterized by repeating tetrafucose units (-l-Fuc-α-1,3-l-Fuc2*S*-α-1,3-l-Fuc2*S*-α-1,3-l-Fuc2*S*4*S*-α-1,3)_n_ [88]. A structural activity study of the sea urchins *Strongylocentrotus franciscanus*, *Lytechinus variegatus* and *Echinometra lucunter* with the derived sulfated polysaccharides 2-sulfated SF, 4-sulfated SF and 2-sulfated SG, respectively, provided a good model to systematically relate the positional effect of sulfation, nature of sugar and molecular weight with the anticoagulant properties of these glycans. The study revealed that 4-sulfated SF and 2-sulfated SG exhibited a potent anticoagulant effect when examined in vitro and in vivo. The 2-sulfated SG, however, emerged to be the most potent, while the 2-sulfated SF did not exhibit any anticoagulation effect. This suggested a plausible role for the nature of sugar on the observed activity, as shown in Table 6. The study reaffirms the role of molecular weight and sulfation pattern to be critical in antithrombotic and anticoagulant activity, as observed in many previous studies [89].

##### Other Sulfated Polysaccharides

Alginate

Brown algal organic polymers called alginates are established to have therapeutic applications in cases of cardiovascular disease. The sulfation of alginate leads to the formation of propylene glycol alginate sodium sulfate (PSS) (Figure 12), which has been shown to be a potent anticoagulant. Its similarity with the structure of heparin is reflected in its similar anticoagulant activity. However, PSS displayed less aPTT and TT prolongation than heparin. When PSS was subjected to hydrolysis, 13 oligosaccharides were obtained. Similar to observations in the case of sulfated polysaccharides, the study demonstrated the requirement of a higher degree of sulfation and high molecular weight to display anticoagulant activity; this was detected by in vitro coagulation assays [91].

##### Sulfated Rhamnan

Marine algae have been studied extensively [31], as they are an interesting source of antithrombotic sulfated polysaccharides. Along with SFs and SGs, algae are also a source of sulfated rhamnans. The marine green alga *Monostroma nitidum* is rich in rhamnan polysaccharides. Glycans, recently isolated from this alga, exhibit a structure comprising (-l-Rhap-α-1,3-)_n_ and (-l-Rhap-α-1,2-)_n_ residues with branches consisting of 4-linked β-d-xylose, 4-/6-linked β-d-glucose, terminal β-d-GlcA, and 3-/2-linked α-l-rhamnose. The anticoagulant activity of this rhamnan is caused by the inhibition of FIIa mediated by HC-II and AT. Polysaccharide is also involved with the inhibition of thrombin and FXa via AT [92]. Sulfated heterorhamnans were also isolated from the green marine alga *Gayralia oxysperma* (Go3), with the intention to study their inhibitory effects on venom from *Bothrops jararaca* and *Lachesis muta*. The sulfated heterorhamnans from Go3 were found to be anticoagulants, as they could block the coagulation of plasma in mice which was caused by the venom [93]. Another rhamnan-type sulfated polysaccharide, PF2 (Figure 13), was also purified from the green seaweed *Monostroma angicava*.

It consisted of (-l-Rhap-α-1,3-)_n_ and (-l-Rhap-α-1,2-)_n_ residues and branches at C-2 of (-l-Rhap-α-1,3-)_n_ residues. Sulfation was present in groups at C-3 of rhamnan units. The sulfated polysaccharide PF2 was also found to exhibit high anticoagulant activity [94].

##### Sulfated Chitosan

Chitin is one of the most abundant marine polysaccharides found predominantly in the exoskeleton of arthropods and in fungal cell walls [95]. It is comprised of repeating β-d-GlcNAc units and, due to its nontoxic and biocompatible properties, it was shown to possess several pharmaceutical applications, e.g., in tissue engineering [95]. The insolubility issues related to chitin can be overcome by its partial deacetylation, leading to the formation of chitosan. Chitosan, comprising of β-d-GlcNAc and β-d-glucosamine repeating units, also has many therapeutically significant properties [95]. The sulfation of chitosan, yielding chitosan sulfate, has been shown to exhibit anticoagulant activity as well. For example, chitosan sulfate derived from a squid *Doryteuthis singhalensis* showed anticoagulant activity, with an aPTT of 6.91 IU/mg and a PT of 1.85 IU/mg [96]. A low molecular weight sulfated chitosan obtained from the cuttlebone of *Sepia pharaonis* (Figure 14), when assayed against avian blood for anticoagulant activity, showed a prolongation of aPTT and PT. The observed aPTT and PT of 66.7 sec and 94.8 sec, respectively, by this sulfated chitosan was even higher than that exhibited by a standard heparin control [97]. A *Sepia prashadi*-derived sulfated chitosan also exhibited a potent anticoagulant function, with an aPTT value of 6.90IU/mg and a PT value of 1.2 IU/mg units. Sulfation has been considered an important factor in conferring anticoagulant properties in chitosans [98].

## 3. Concluding Remarks

The studies reported on marine antithrombotics strongly encourage the development of such unique molecules as future antithrombotics. These molecules, by their distinct structure, unique chemical and physical properties, exhibit different anticoagulant/antithrombotic activities. These activities are correlated with their structural features such as molecular weight, glycosidic linkage, sulfate content and pattern, substitution, charge density, etc. A combination of these factors, on the one hand makes these compounds highly unique, but on the other, makes them susceptible to certain limitations as well.

For example, peptides, due to their small size, are considered significant future therapeutics. However, their serum stability and short half-lives are some of the challenges that need to be overcome. Similarly, the large molecular weight of sulfated glycans was found to be crucial for their antithrombotic activity but has a downside from a pharmaceutical perspective. The associated heterogeneity of these molecules poses further complications for their development as potential antithrombotics. The high viscosity of sugars leads to practical challenges associated with the mode of administration. Observations with some of the polysaccharides isolated from sea cucumber, which showed that they exhibit a strong tendency for platelet aggregation and FXII activation, are a significant concern. However, there remains a lot of focus on these molecules, specifically sulfated glycans. This is because some of these molecules have been found to exhibit anticoagulant potential like heparin yet with fewer side effects. A few SFs have been shown to prevent clotting while not activating FXII or causing platelet aggregation. One of the major advantages with marine sulfated glycans is that because they are not from vertebrates, their oral administration does not lead to degradation due to the absence of digestive enzymes. This could be advantageous for oral administration of these antithrombotic molecules and is a crucial point, as one of the major downsides of heparin is its subcutaneous or intravenous mode of administration and limited bioavailability. However, this also raises an obvious concern regarding renal toxicity issues due to the non-systemic clearance of the drug from the body.

There are many factors that modulate the activity of marine molecules and need to be understood. Improved technical developments have led to easier isolation and characterization of marine compounds, which has made their structural elucidation easier, paving the way for a better understanding of the structure–activity relationship. The studies reviewed aimed to identify antithrombotics from various marine sources but there still needs to be detailed biological investigations to establish their clinical prospects. Nonetheless, these reports are significant in providing a repository of potential molecules which could be screened or rationally improved for development as future antithrombotics. Moreover, their structural characterization and mechanistic studies will be insightful in designing synthetic mimetics using theses natural marine molecules as a template.

## Figures and Tables

**Figure 1 marinedrugs-18-00514-f001:**
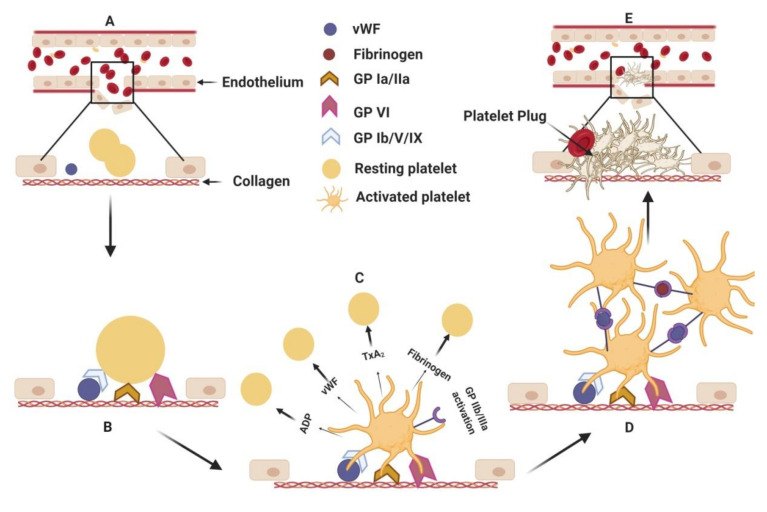
Adhesion, activation and aggregation. (**A**) Endothelial injury in the blood vessel exposes collagen to platelets and von Willebrand factors (vWF). (**B**) Resting platelets attach to the exposed collagen with their glycoprotein surface receptors directly and/or via vWF mediated anchorage. (**C**) The platelet, upon adhesion to collagen, is activated, changes morphology and releases granular contents which activate more platelets. Activated platelets induce conformational change in a surface receptor, namely, glycoprotein receptor (GP)IIb/IIIa, which induces (**D**) platelet aggregation via vWF and fibrinogen. (**E**) Aggregated platelets form a plug at the site of the injury, thereby stopping the blood leakage temporarily. ADP: Adenosine diphosphate; TxA_2_: Thromboxane A_2_; vWF: von Willebrand factor; GP: glycoprotein receptors. (Figure courtesy of biorender.com).

**Figure 2 marinedrugs-18-00514-f002:**
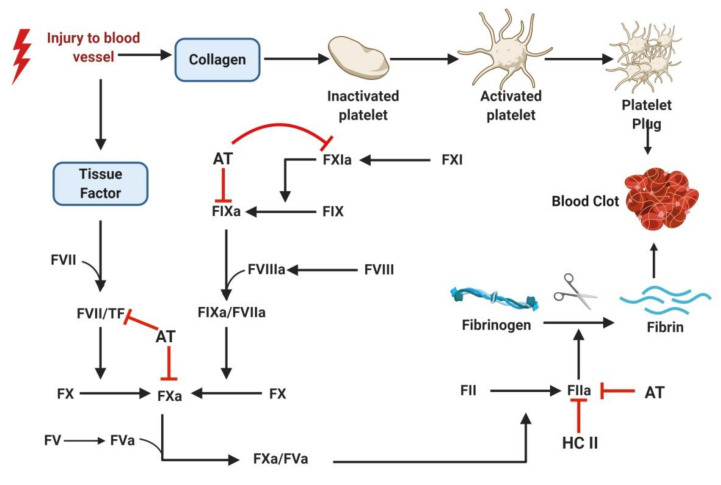
Coagulation cascade depicting blood clot formation. Exposure of tissue factor and subendothelial collagen post blood vessel injury triggers platelet aggregation and blood clot formation. Coagulation cascade involves conversion of inactive coagulation factor to their corresponding active form by a series of enzymatic reactions. The cascade concludes with the conversion of fibrinogen to fibrin, forming a mesh-like network with entangled blood cells and creating the blood clot. Coagulation factors are represented by Roman numerals, e.g., prothrombin is factor II (FII). When zymogens (inactivated proteases) are activated, the letter “a” is presented after the Roman numeral, e.g., thrombin is factor IIa (FIIa). As conventionally used, procoagulant and anticoagulant pathways are indicated in black and red, respectively. AT: antithrombin; HC II: Heparin cofactor II; TF: Tissue factor. (Figure courtesy of biorender.com).

**Figure 3 marinedrugs-18-00514-f003:**
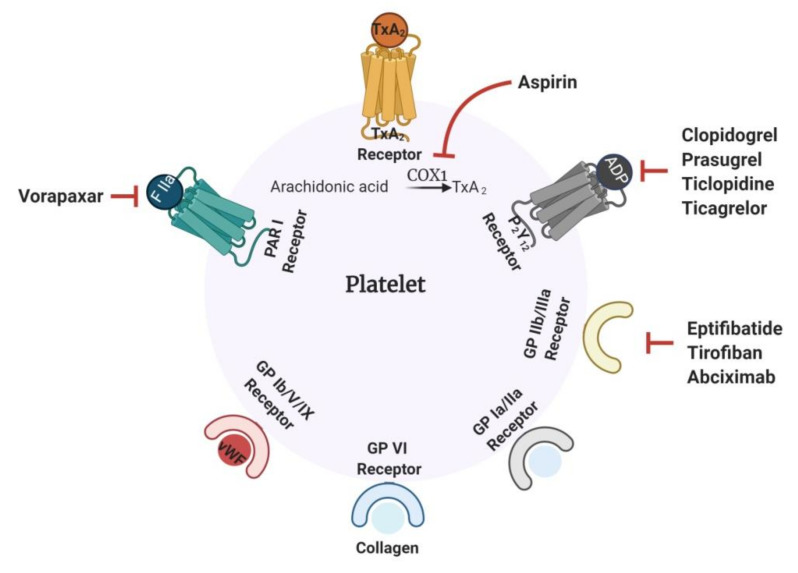
Mechanism of action of common antiplatelet drugs. Platelet surface receptors play a major role in activation, adhesion and aggregation events. Drugs targeting the receptors—PAR 1, P_2_Y_12_ and GP IIb/IIIa receptors—have potent antiplatelet implications. Some of the drugs targeting receptors are shown in the figure. Aspirin, a widely prescribed antiplatelet drug, acts by inhibiting the COX 1 enzyme required for the conversion of arachidonic acid to TxA_2_.PAR 1: protease-activated receptor 1; P_2_Y_12_: purinergic G protein-coupled receptor; COX 1: cyclooxygenase 1; TxA_2_: thromboxane A_2_. (Figure courtesy of biorender.com).

**Figure 4 marinedrugs-18-00514-f004:**
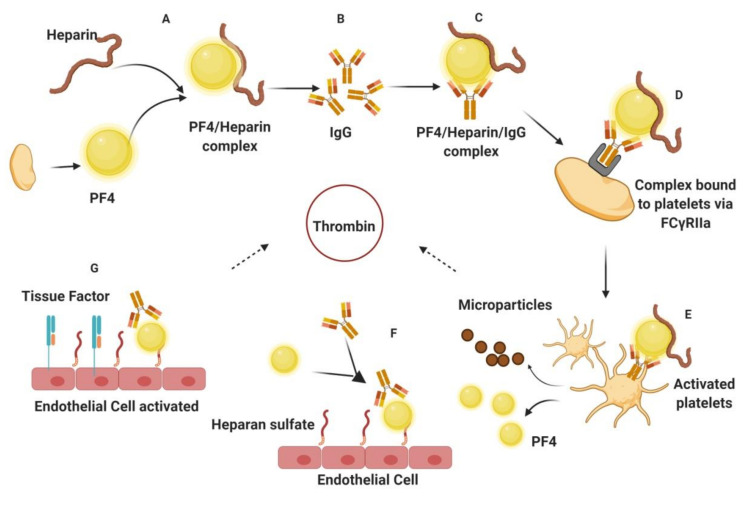
Pathophysiology of heparin-induced thrombocytopenia (HIT). (**A**) PF 4, a small molecule released from platelets, binds to heparin, forming an immunogenic complex, which (**B**) elicits IgG antibody production. (**C**) The formation and presence of IgG triggers the production of the PF 4/Heparin/IgG complex. (**D**) This ternary complex binds to platelets via the FCγRIIa receptor. (**E**) Activated platelets release microparticles and more PF 4, accelerating the system. Microparticle promotes thrombin release. (**F**) IgG/PF 4 complex binds to heparin, thus activating endothelial cells to release (**G**) tissue factor, which consequently enhances thrombin generation. PF 4: Platelet factor 4; IgG: Immunoglobulin G. (Figure courtesy of biorender.com).

**Figure 5 marinedrugs-18-00514-f005:**
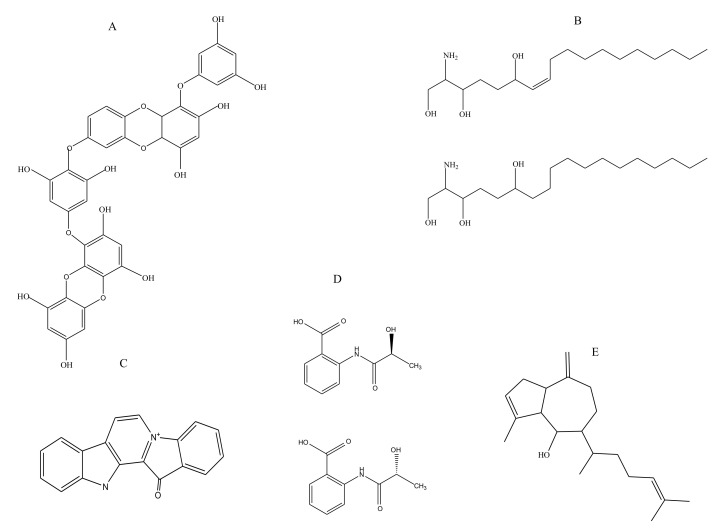
Representative structures of some small marine molecules endowed with antithrombotic action: (**A**) polyphenol (dieckol) isolated from *Eisenia bicyclis*, (**B**) sphingosines isolated from marine sponge *Haliclona tubifera*, (**C**) alkaloid (fascaplysin) isolated from marine sponge *Fascaplysin opsis*, (**D**) benzoic acid derivatives isolated from marine fungus *Penicillium chrysogenum*, and (**E**) Triterpenoid (Pachydichtyol A) isolated from marine brown alga *Dictyota menstrualis*.

**Figure 6 marinedrugs-18-00514-f006:**
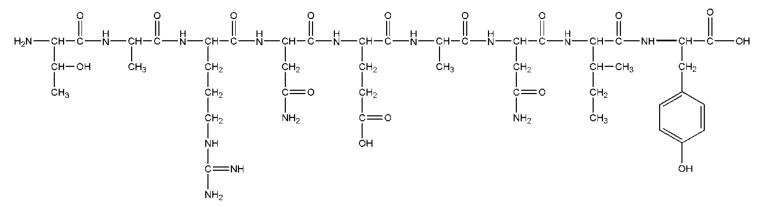
Structure of marine peptide CAGP, isolated from marine oyster *Crassostrea gigas*, which was identified to be the prime component of the anticoagulant fraction in the oyster extract. The sequence of this peptide was identified to be TARNEANVNIY.

**Figure 7 marinedrugs-18-00514-f007:**
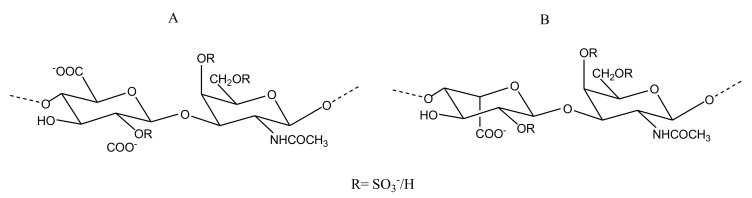
Repeating disaccharide structural unit of CS/DS (chondroitin sulfate/dermatan sulfate) isolated from the skin of the fish *Sciaena umbra*. The structure comprises of repeating units of uronic acid and *N*-acetyl β-d-galactosamine linked by 4- and 3-linkages, respectively. CS (**A**) has β-d-glucuronic acid while DS (**B**) has α-l-iduronic acid residues.

**Figure 8 marinedrugs-18-00514-f008:**
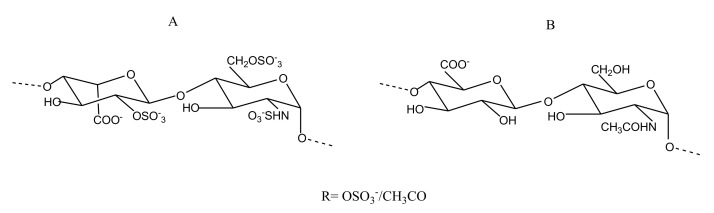
Structure of Hp (heparin) (**A**) and HS (heparan sulfate) (**B**) reported from the shrimp *Litopenaeus vannamei*. The Hp structure consists of repeating units of predominantly α-d-iduronic acid 4-linked to N-acetyl α-d-glucosamine, also 4-linked. In HS, the uronic acid unit is mostly β-d-glucuronic acid.

**Figure 9 marinedrugs-18-00514-f009:**
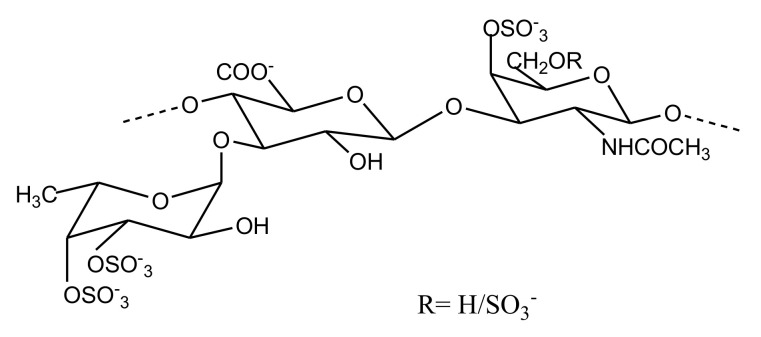
FucCS (fucosylated chondroitin sulfates) isolated from sea cucumber *Massinium magnum*. Structure comprises of β-d-glucuronic acid and *N*-acetyl β-d-galactosamine units linked via β-1,4 linkages, while sulfated α-l-fucopyranosyl branches are linked to β-d-glucuronic acid by α-1,3 linkage.

**Figure 10 marinedrugs-18-00514-f010:**
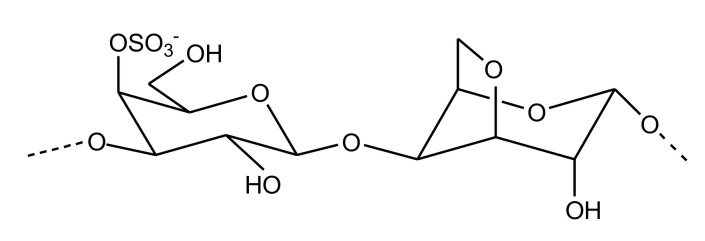
Structure of the sulfated galactan isolated from red seaweed *Spyridia hypnoides*. Its *s*tructure comprises a β-d-galactopyranose unit linked to 3, 6-anhydro-α-l-galactopyranose unit via β-1,4 linkages.

**Figure 11 marinedrugs-18-00514-f011:**
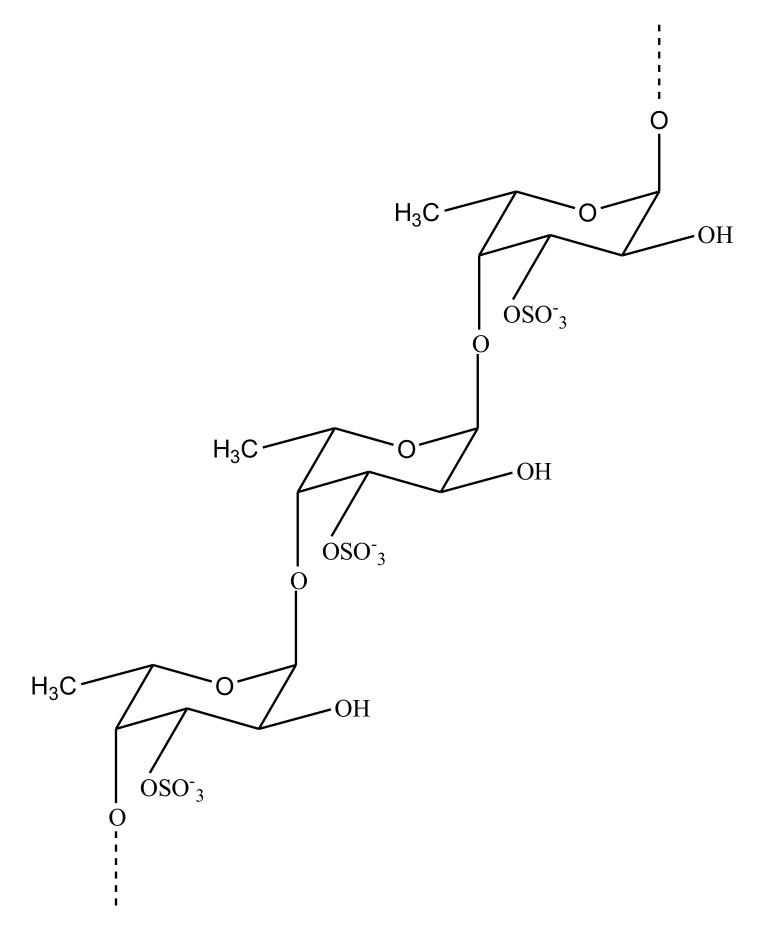
Structure of the sulfated fucan (SF) isolated from sea cucumber *Holothuria fuscopunctata* comprising l-fucose repeating units linked by α-1,3 linkages.

**Figure 12 marinedrugs-18-00514-f012:**
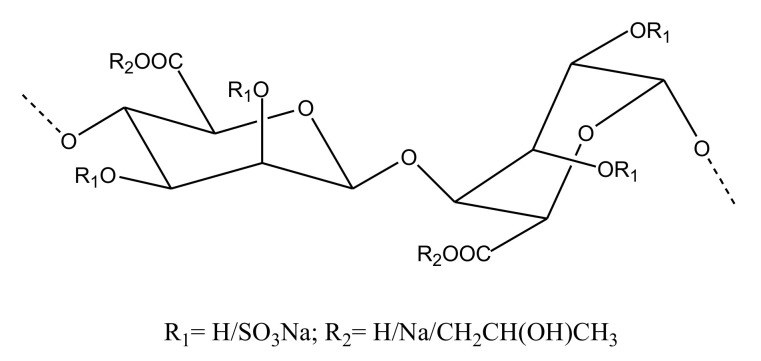
Representative structure of propylene glycol alginate sodium sulfate extracted from a brown alga.

**Figure 13 marinedrugs-18-00514-f013:**
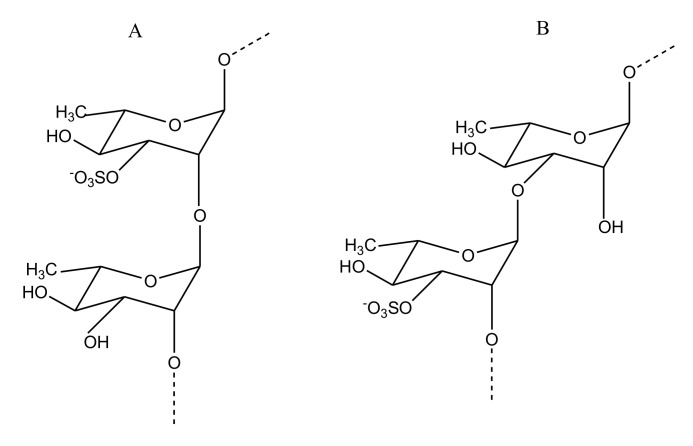
Structure showing repeating units of sulfated rhamnan isolated from green seaweed *Monostroma angicava*. Repeating units comprise of (**A**) α-1,2 linked and (**B**) α-1,3 linked sulfated rhamnans.

**Figure 14 marinedrugs-18-00514-f014:**
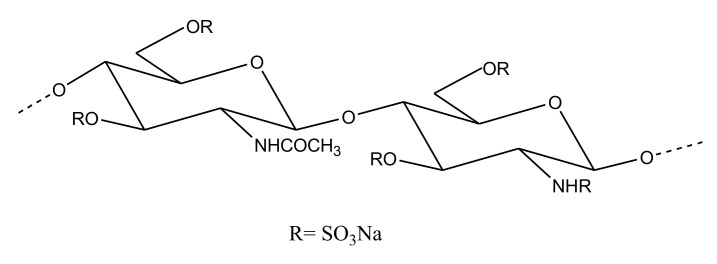
Structure representing repeating unit of sulfated chitosan derived from cuttlefish *Sepia pharaonis.*

**Table 1 marinedrugs-18-00514-t001:** Antithrombotic activity data of the R/S HPABA ((2-Hydroxypropanamido) benzoic acid) in vivo model of rats.

Group	Dose (mg/kg)	Bleeding Time (min) ^a^	Clotting Time (sec) ^a^	Thrombus ^b^ Weight (mg)	Recovery ^c^ Number
*R*-HPABA	100	11.04 ± 0.76	247 ± 26.1	1.90 ± 0.10	6
*S*-HPABA	100	9.50 ± 0.60	230 ± 24.4	2.14 ± 0.18	3
Aspirin	100	10.1 ± 0.35	244 ± 24.7	2.05 ± 0.17	4

^a^ Result after oral administration of HPABA, aspirin in mice after 7 days.^b^ Result of HPABA, aspirin on common carotid artery thrombosis in rats.^c^ Result of HPABA, aspirin on collagen—epinephrine induced pulmonary thrombosis in mice after 7days (*n* = 10).

**Table 2 marinedrugs-18-00514-t002:** Inhibitory activity of polar lipids extracted from *Salmon salar* against PAF (platelet activating factor) and thrombin.

Total Polar Lipids (*Salamon salar)*	PAF Inhibition IC_50_ (μg)	Thrombin Inhibition IC_50_ (μg)
**Conventional extraction**	45 ± 22	382 ± 39
**Food grade extraction**	86 ± 18	102 ± 29

**Table 3 marinedrugs-18-00514-t003:** Anticoagulant activity of different protein hydrolysates containing digested peptides of oyster muscle.

Activity (sec)	Concentration (mg/mL)	Native Control	Trypsin	Papain	Neutrase	Pepsin
**aPTT**	50	41.7 ± 0.53	46.1 ± 1.04	41.6 ± 0.42	60.4 ± 1.87	183.8 ± 10.26
**TT**	50	9.9 ± 0.12	21.4 ± 0.29	14.7 ± 0.35	33.8 ± 2.56	31.8 ± 0.56

aPTT—activated partial thromboplastin time; TT—thrombin time.

**Table 4 marinedrugs-18-00514-t004:** Values of HS release from endothelial cells post-stimulated with sH/HS from the shrimp *Litopenaeus vannamei*. Values are in cpm/μg of protein.

Dose (μg/mL)	Hp/HS	Heparin	Control
**100**	40,000	40,000	15,000
**50**	35,000	-	-
**25**	30,000	-	-

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
