# Peer review of "Marine Antithrombotics"

_marinedrugs, 2020, doi:10.3390/md18100514_

Round 1

Reviewer 1 Report

The review summarizes the available marine molecules with interest in as antithrombotic drugs. The introduction is very didactic and useful to understand the complex pathways of coagulation. Also the concluding remarks are useful and critical to have a general idea of what molecules could be more useful for the future.

I want to give only some minor details/suggestions to address before publication:

In the introduction I suggest to add a disadvantage of the direct oral anticoagulants (DOACs) that is the few antidotes available and its expensive nature. For the DOACs approved by the regulation agencies in USA and Europe just two antidotes are available: idarucizumab and adexanet.  Both molecules are recombinant proteins with a very high cost. In addition, clinical experience with them is rather limited and the safety profile is still under debate (https://www.thieme-connect.de/products/ejournals/abstract/10.1160/TH14-11-0982  http://www.sciencedirect.com/science/article/pii/S1053077018306050) . This is especially relevant for the use of anticoagulants during cardiac surgeries and for patiens under renal failure and the cause of the still extended use of high molecular weight heparin that has a cheap and secure antidote (protamine).

The term mimetic means “representing or imitating something “according to the Cambridge dictionary. In this and in my opinion the term GAG-mimetic is valid for those synthetic systems where chemists try to imitate the structure of naturally occurring GAGs. However, the Sulphated Fucans and Sulfated galactans are naturally occurring molecules. In this sense I believe that should be rather call GAG-like molecules than GAG-mimetics.

Considering the molecules included in the manuscript under GAG mimetic I would only include chemically sulphated Alginate. However, I believe that non-natural molecules could be not be included in this review. I fact the only one included is the sulphated alginate. I left this on the authors decisdion but ff the authors prefer to include a section of GAG-mimetic with chemically sulphated marine derived polysaccharides such sulphated alginate they should add sulphated chitin and chitosan. I put below some papers on anticoagulant activity of sulphated chitosan but there might be  more:

https://www.sciencedirect.com/science/article/abs/pii/000862158485380X?via%3Dihub

https://www.sciencedirect.com/science/article/pii/S0008621502000988?via%3Dihub

https://www.sciencedirect.com/science/article/pii/S0141813012000098?via%3Dihub

I also know and example of a sulfated marine Bacterial polysaccharide with anticoagulant properties that is shoul be included under other sulphated polysaccharides:

https://www.sciencedirect.com/science/article/pii/S0304416501001854?via%3Dihub

Other detail:

 “ Sulfated polysaccharides found in marine organisms include 396 glycosaminoglycans (GAGs) like, chondroitin sulfate (CS), dermatan sulfate (DS), heparin, heparan sulfate 397 (HS), fucosylated chondroitin sulfates (FucCS), and GAG mimetics like sulfated fucans (SFs), sulfated 398 galactans (SGs) as well as other polysaccharide like alginates, rhamnan, chitins etc..”

The sentence included in the manuscript appear to say that chitin is sulphated. This was, for sure ,not the intention of the authors. It should be rewritten.

Author Response

Reviewer #1:

Comments and Suggestions for Authors

The review summarizes the available marine molecules with interest in as antithrombotic drugs. The introduction is very didactic and useful to understand the complex pathways of coagulation. Also the concluding remarks are useful and critical to have a general idea of what molecules could be more useful for the future.

I want to give only some minor details/suggestions to address before publication:

In the introduction I suggest to add a disadvantage of the direct oral anticoagulants (DOACs) that is the few antidotes available and its expensive nature. For the DOACs approved by the regulation agencies in USA and Europe just two antidotes are available: idarucizumab and adexanet.  Both molecules are recombinant proteins with a very high cost. In addition, clinical experience with them is rather limited and the safety profile is still under debate (https://www.thieme-connect.de/products/ejournals/abstract/10.1160/TH14-11-0982 http://www.sciencedirect.com/science/article/pii/S1053077018306050). This is especially relevant for the use of anticoagulants during cardiac surgeries and for patiens under renal failure and the cause of the still extended use of high molecular weight heparin that has a cheap and secure antidote (protamine).

Answer: We have added the comment addressing the specific disadvantages of DOACs as suggested by the reviewer in terms of limited availability of inexpensive and safe reversal drugs. Suggested and other appropriate references have been cited.

The term mimetic means “representing or imitating something “according to the Cambridge dictionary. In this and in my opinion the term GAG-mimetic is valid for those synthetic systems where chemists try to imitate the structure of naturally occurring GAGs. However, the Sulphated Fucans and Sulfated galactans are naturally occurring molecules. In this sense I believe that should be rather call GAG-like molecules than GAG-mimetics.

Answer: We have corrected the terminology throughout the paper, changing GAG mimetics to GAG-like molecules, as suggested.

Considering the molecules included in the manuscript under GAG mimetic I would only include chemically sulphated Alginate. However, I believe that non-natural molecules could be not be included in this review. I fact the only one included is the sulphated alginate. I left this on the authors decisdion but ff the authors prefer to include a section of GAG-mimetic with chemically sulphated marine derived polysaccharides such sulphated alginate they should add sulphated chitin and chitosan. I put below some papers on anticoagulant activity of sulphated chitosan but there might be  more:

https://www.sciencedirect.com/science/article/abs/pii/000862158485380X?via%3Dihub

https://www.sciencedirect.com/science/article/pii/S0008621502000988?via%3Dihub

https://www.sciencedirect.com/science/article/pii/S0141813012000098?via%3Dihub

I also know and example of a sulfated marine Bacterial polysaccharide with anticoagulant properties that is shoul be included under other sulphated polysaccharides:

https://www.sciencedirect.com/science/article/pii/S0304416501001854?via%3Dihub

Answer: We decided to include all these molecules in our paper since they generally fall somewhat inside the scope of marine antithrombotics, either chemically modified at the marine precursor or entirely of natural source. There are indeed studies showing anticoagulant activity of derivatized chitosan. As, we are reviewing the articles published in last five years, we did not include the aforementioned papers. However, we did include three studies reporting anticoagulant activity of sulfated chitosan from marine species from recent publications.

Other detail:

 “ Sulfated polysaccharides found in marine organisms include 396 glycosaminoglycans (GAGs) like, chondroitin sulfate (CS), dermatan sulfate (DS), heparin, heparan sulfate 397 (HS), fucosylated chondroitin sulfates (FucCS), and GAG mimetics like sulfated fucans (SFs), sulfated 398 galactans (SGs) as well as other polysaccharide like alginates, rhamnan, chitins etc..”

The sentence included in the manuscript appear to say that chitin is sulphated. This was, for sure, not the intention of the authors. It should be rewritten.

Answer: We have corrected this sentence to avoid the message that chitins are sulfated.

Comments from authors: We want to acknowledge your comments regarding our submission. They have helped us to improve the quality of our presentation during the revision process.

Reviewer 2 Report

The manuscript entitled “Marine Antithrombotics” by Dwivedi & Pomin presents the current knowledge on the chemical composition and structural characteristics of marine molecules and correlates these parameters with their antithrombotic /anticoagulant properties. The topic is interesting and the manuscript is well organized and written.

Yet, the theme has been recently covered by the article of F. Carvalhal, R. R. Cristelo, D. I. S. P. Resende, M. M. M. Pinto, E. Sousa and M. Correia-da-Silva, entitled “Antithrombotics from the Sea: Polysaccharides and Beyond”, and it was published in Marine Drugs in 2019 (Mar. Drugs 2019, 17, 170; doi:10.3390/md17030170).

Therefore, the authors should present to the editor the novelty in their manuscript.

General comment:

Write anti-FXa (or anti FXa) etc in a consistent way

Specific points

Please correct the following points

Line 54, 55: fibrinogen is written twice, delete one

Line 63, Figure 1 legend: Adhesion (A in capital letters).

Line 95: determine

Line 117: purinergic G protein-coupled receptor

Line 178: Please correct KATP

Line 125: of the oldest

Line 148: parenteral mode of administration

Line 153: form an immunogenic

Line 208: and alkaloids

Line 213: components

Line 214: (Figure 5A)

Line 223: can decrease

Line 232: derivatives

Figure 5: The terpene structures from the text could be added

Line 257: (PI3K)

Line 261: while increased

Line 263: (Figure 5D)

Line 264: properties

Line 263: (Figure 5D)

Line 270: to be somewhat correlated with its inhibition COX 1 activity [36].

Line 275: Alkaloids

Line 278: (Figure 5C)

Line 279: PI3K pathway

Line 287: Phospholipids (PL)

Table 2: Total polar lipids

Line 299: ω3 PUFAs were found to be predominant in the salmon PL preparation and their content was higher than that of omega 6 (ω6) fatty acids

Line 304: Interestingly, FGE extracted

Line 325: investigations

Line 327: a widely

Line 355: were

Line 360: [46].

Line 381: PI3K

Line 388: in in vitro analysis

Line 399: across the species of seaweeds, fishes, sea cucumbers, marine algae, shrimps, sea urchins, fishes etc.

Figure 8: The R groups are not shown in the structures

Line 496: 10-fold higher anti-FXase

Line 498: were less than the native sugar, signifying the effect

Line 507: Stichopus hermanni (in italics)

Lines 574-577: Delete “backbone…[59].

Figure 12: Correct the structure, saccharide bonds to C (not to R2)

Line 636, Figure 13: Repeating units comprise of (A) α-1,2 linked and (B) α-1,3 linked sulfated  rhamnans.

Line 640: Go3, were found anticoagulant

Author Response

Reviewer #2:

Comments and Suggestions for Authors

The manuscript entitled “Marine Antithrombotics” by Dwivedi & Pomin presents the current knowledge on the chemical composition and structural characteristics of marine molecules and correlates these parameters with their antithrombotic /anticoagulant properties. The topic is interesting and the manuscript is well organized and written.

Yet, the theme has been recently covered by the article of F. Carvalhal, R. R. Cristelo, D. I. S. P. Resende, M. M. M. Pinto, E. Sousa and M. Correia-da-Silva, entitled “Antithrombotics from the Sea: Polysaccharides and Beyond”, and it was published in Marine Drugs in 2019 (Mar. Drugs 2019, 17, 170; doi:10.3390/md17030170).

Therefore, the authors should present to the editor the novelty in their manuscript.

Answer: This was also the comment of the editor in charge of handling our submission. As per requested, we are now citing this reference and stating in Introduction, the differences between our current submission and this reference.

General comment:

Write anti-FXa (or anti FXa) etc in a consistent way

Answer: We have revised the use of anti-FXa throughout the entire paper.

Specific points

Please correct the following points

Line 54, 55: fibrinogen is written twice, delete one

Line 63, Figure 1 legend: Adhesion (A in capital letters).

Line 95: determine

Line 117: purinergic G protein-coupled receptor

Line 178: Please correct KATP

Line 125: of the oldest

Line 148: parenteral mode of administration

Line 153: form an immunogenic

Line 208: and alkaloids

Line 213: components

Line 214: (Figure 5A)

Line 223: can decrease

Line 232: derivatives

Figure 5: The terpene structures from the text could be added

Line 257: (PI3K)

Line 261: while increased

Line 263: (Figure 5D)

Line 264: properties

Line 263: (Figure 5D)

Line 270: to be somewhat correlated with its inhibition COX 1 activity [36].

Line 275: Alkaloids

Line 278: (Figure 5C)

Line 279: PI3K pathway

Line 287: Phospholipids (PL)

Table 2: Total polar lipids

Line 299: ω3 PUFAs were found to be predominant in the salmon PL preparation and their content was higher than that of omega 6 (ω6) fatty acids

Line 304: Interestingly, FGE extracted

Line 325: investigations

Line 327: a widely

Line 355: were

Line 360: [46].

Line 381: PI3K

Line 388: in in vitro analysis

Line 399: across the species of seaweeds, fishes, sea cucumbers, marine algae, shrimps, sea urchins, fishes etc.

Figure 8: The R groups are not shown in the structures

Line 496: 10-fold higher anti-FXase

Line 498: were less than the native sugar, signifying the effect

Line 507: Stichopus hermanni (in italics)

Lines 574-577: Delete “backbone…[59].

Figure 12: Correct the structure, saccharide bonds to C (not to R2)

Line 636, Figure 13: Repeating units comprise of (A) α-1,2 linked and (B) α-1,3 linked sulfated  rhamnans.

Line 640: Go3, were found anticoagulant

Answer: Thanks for your comments on all these points. We have fixed all of them, with no exception. We want to acknowledge your comments regarding our submission and thank for your great and deep review. They have helped us to improve the quality of our presentation during the revision process.

Reviewer 3 Report

The author provide a comprehensive review on marine antithrombotics/ The article is well illustsated. Varios classes of substances with anticoagulant activity have been describe. Considerating the impotant role of hypercoagulationin the COVID-19 pathogenesis, it is recomended to assess prospects for use these  substances  in complex therapy of intoxication caused by Cov-2.

Author Response

Reviewer #3:

Comments and Suggestions for Authors

The author provide a comprehensive review on marine antithrombotics/ The article is well illustsated. Varios classes of substances with anticoagulant activity have been describe. Considerating the impotant role of hypercoagulation in the COVID-19 pathogenesis, it is recomended to assess prospects for use these substances in complex therapy of intoxication caused by Cov-2.

Answer: Thanks for your positive comments regarding our submission and support to our publication.